# Anti-Apoptotic Effect of Chrysophanol Isolated from *Cassia tora* Seed Extract on Blue-Light-Induced A2E-Loaded Human Retinal Pigment Epithelial Cells

**DOI:** 10.3390/ijms24076676

**Published:** 2023-04-03

**Authors:** Su-Kang Kim, Ju-Yeon Ban, Hyungoo Kang, Sang-il Park

**Affiliations:** 1Department of Biomedical Laboratory Science, Catholic Kwandong University, Gangneung-si 25601, Republic of Korea; 2Department of Dental Pharmacology, College of Dentistry, Dankook University, Cheonan-si 31116, Republic of Korea; 3Department of Optometry, Catholic Kwandong University, Gangneung-si 25601, Republic of Korea

**Keywords:** retinal pigment epithelium, blue light, apoptosis, chrysophanol

## Abstract

The seeds of *Cassia tora* (*C. tora*) species mainly contain anthraquinone, anthraquinone glycoside, and naphthalene derivatives. We investigated the anti-apoptotic effects of *C. tora* seed extract and its isolated compounds on blue-light-induced lipofuscin (A2E)-loaded human retinal pigment epithelial (RPE) cells. For analysis of the *C. tora* extract, high-performance liquid chromatography method was used. A2E-loaded human retinal pigment epithelial cells and blue light were used to create excessive photo-oxidation to induce cell death. Lactate dehydrogenase (LDH) assay was used to measure cell cytotoxicity, and the mRNA expression of genes involved in apoptosis was examined to evaluate the mechanism of cell death. *C. tora* extract, *n*-hexane fraction, and chrysophanol were found to inhibit apoptotic cell death. Additionally, *C. tora* extract, *n*-hexane fraction, and chrysophanol reduced the mRNA expression of genes involved in the apoptosis pathway. *C. tora* and chrysophanol were considered to inhibit apoptosis and oxidative stress response. The major component of *C. tora* has a protective effect against apoptosis. The ingredients of *C. tora* can be used as therapeutic substances or to prevent diseases caused by the excessive oxidation of A2E substances in the retina, such as in age-related macular degeneration.

## 1. Introduction

The retina is the innermost layer in the eye and plays an important role in the visual processing that turns light energy from photons into three-dimensional images [1]. Its cells are divided into three basic cell types, including photoreceptor, neuronal, and glial cells. Additionally, the retina is a layered structure, with 10 distinct layers of neurons interconnected by synapses [2], including the inner limiting membrane (ILM), outer limiting membrane (OLM), nerve fiber layer (NFL), ganglion cell layer (GCL), inner plexiform layer, outer plexiform layer (OPL), inner nuclear layer (INL), outer nuclear layer (ONL), photoreceptor layer (PL), and retinal pigmented epithelium (RPE) monolayer [3]. Among these layers, RPE cells perform important functions in the visual process, and thus the dysfunction of RPE cells may contribute to the development of various retinal diseases, such as age-related macular degeneration (AMD).

RPE cells regulate the transport of nutrients and waste products in the retina, promoting outer segment renewal by ingesting and destroying the used end of the outer segment of photoreceptor cells [4]. They also protect the retina externally from excessive high energy radiation and photogenic forms that react with oxygen and maintain retinal homeostasis by releasing diffusion factors [5]. 

AMD is the leading cause of irreversible blindness among the elderly [6]. AMD is commonly characterized by progressive macular degeneration of RPE cells and photoreceptors, lipofuscin (A2E) accumulation, and drusen formation [7]. Oxidative stress/damage to RPE cells caused by excessive exposure to light contributes to the development of AMD [8]. When exposed to blue light, A2E not only generates reactive oxygen species but also photo-oxidizes along its conjugated double bonds to yield more toxic products [9,10]. Therefore, the photo-oxidized forms of A2E are highly reactive and likely contribute to RPE cell damage and death. Blue-light-induced A2E-loaded RPE cell DNA damage induces apoptosis by mechanisms that require ABL1, TP53, and the stress kinase pathway [11,12]. 

*Cassia tora* (*C. tora*) is an annual herb that is widely grown in east Asia, northern Australia, and the Americas [13]. In ancient systems of medicine, such as Ayurveda, *C. tora* was used to treat a variety of medical complications, including bronchitis, constipation, conjunctivitis, ulcers, hypertension, hypercholesterolemia, liver damage, fungal infection, diabetes, edema, glaucoma, nyctalopia, ringworm, and skin diseases [14]. The seed of *C. tora* species mainly contains anthraquinone, anthraquinone glycoside, and naphthalene derivatives [15,16,17]. Regarding anthraquinone and its glycoside, hundreds of in vitro and animal studies have reported anti-inflammatory, antioxidant, antifungal, and antiviral effects. In addition, *C. tora* seed extract was shown to have hypolipidemic activity [18]. In the study by Ko et al., results confirmed the effects of *C. tora* extracts emodin and reain on retinal diseases induced by type 2 diabetes. As such, it has been reported that *C. tora* extract and its compounds are effective for retinal diseases [11]. A recent study demonstrated that exposure to IR radiation induces adipogenesis and sebum production through activation of JNK/p38 MAPK signaling in human sebaceous cells and that these effects are mitigated by chrysophanol [19]. Through the above study, it was confirmed that *C. tora* seed has various pharmacological activities. 

In this study, we investigated the efficacies of 70% ethanol *C. tora* seed extracts and its isolated compounds in preventing blue-light-induced A2E-containing RPE cell death as well their role in the possible apoptosis mechanisms involved in cell survival. Our study suggests that *C. tora* extract and its major compound have an anti-apoptotic effect.

## 2. Results

### 2.1. Structure Determination of Anti-Apoptotic Effect Compound Isolation from C. tora Extraction

The extracted seed of *C. tora* was evaporated until dry under reduced pressure to produce 70% ethanol extract (57 g; yield 14.2%). *C. tora* was analyzed using HPLC. Figure 1 shows the *n*-hexane fraction as 2.79 g (yield, 4.89%), EtOAc fraction as 5.49 g (yield, 9.63%), *n*-BuOH fraction as 19.58 g (yield, 34.35%), and the aqueous fraction as 20.56 g (yield, 36.08%). In the fractionation process, we separately fractionated the dichloromethane (DCM) fraction. However, as a result of HPLC analysis of the fractions, there were many overlapping materials with the EtOAc fraction and the *n*-hexane fraction. Therefore, the DCM fraction was excluded from comparison. We obtained two compounds in the *n*-hexane fraction. An isolated compound was obtained as a yellow-colored powder. The molecular formula of one was determined to be C_15_H_12_O_5_ by ESI-MS. The negative ESI-MS of one exhibited a significant peak at m./z 271.08204 [M-H]^+^. The 1H-NMR spectrum revealed signals corresponding to the presence of two prepositioned chelated hydroxyl groups, δ 11.01 (1H, s, 8-OH) and 9.67 (1H, s, 7-OH); three aromatic protons, δ 7.45 (1H, d, J = 7.0, H-9), 6.52 (1H, d, J = 7.7, H-4), 6.48 (1H, s, H-6), and 6.08 (1H, s, H-2); three protons of one methoxyl group at δ 4.01 (3H, s, 5-OCH3) and one methyl group at δ 2.42. The 13C-NMR spectral data showed one carbonyl carbon resonation at δ 162.4; aromatic carbons at δ 55.4, 96.7, 102.7, 102.8, 105.9, 106.6, 106.9, 140.2, 152.5, 161.9, and 168.9; one methoxyl group at δ 55.8; and one methyl group at δ 14.1. By comparing the ESI-MS, 1H-NMR, and 13C-NMR spectral data with that reported in the literature [20], this compound was identified as 5,6-dihydroxy-8-methoxy-2methyl-4H benzo chromen-4-one, which was confirmed as being rubrofusarin. Another isolated compound was obtained as an orange-colored powder (41.1 mg). The molecular formula of the compound was determined to be C_15_H_10_O_4_ by ESI-MS. The negative ESI-MS of the compound exhibited a significant peak at m./z 253.05730 [M-H]^+^. The 1H-NMR spectrum revealed signals corresponding to the presence of two prepositioned chelated hydroxyl groups, δ 11.99 (1H, s, 1-OH) and 12.10 (1H, s, 8-OH); three aromatic protons at 7.09 (1H, s, H-2), 7.28 (1H, d, J = 7.7, H-7), 7.64 (1H, s, H-4), 7.66 (1H, t, J = 7.7, 8.4, H-6), and 7.80 (1H, d, J = 7.0, H-5); and three protons of one methoxyl group at δ 2.46 (3H, s, CH3). The 13C NMR spectral data showed two carbonyl carbon resonations at δ 181.9 and 192.5; aromatic carbons at δ 113.7, 115.9, 119.9, 121.4, 124.4,124.6, 133.3, 133.7, 136.9, and 149.4; and one methyl group at δ 22.3. By comparing the ESI-MS, 1H-NMR, and 13C-NMR spectral data with that reported in the literature, this compound was identified as 3-methyl-1, 8-dihydroxy anthraquinone, which was confirmed to be chrysophanol (Figure 2).

### 2.2. Protective Effect of C. Tora Seed Extract, Its Fractionations, and Chrysophanol on Blue-Light-Induced Cell Death in A2E-Loaded ARPE-19 Cells

It is well known that blue light irradiation causes A2E-loaded ARPE-19 cell death [21]. Prior to the anti-photo-oxidation experiment, to confirm the effect of blue light on A2E-free ARPE-19 cells, blue light was irradiated for up to 20 min, and as a result, cell death was not affected (Figure 3A). A2E induces apoptosis by photo-oxidation, and high concentrations of A2E are toxic by themselves. We confirmed the toxicity of A2E without irradiation with blue light to confirm cell death by photo-oxidation. As a result of confirming A2E toxicity, cell viability was not affected at 20 μm or less (Figure 3B).

In our experiments, we tried to confirm the protective effect of *Cassia tora* extract on RPE cells with accumulated A2E and to identify the most important components. In this process, the activity concentrations of the extract and each fraction were confirmed, and the active compound was sought as the most active *n*-hexane fraction. Cell viability was confirmed by LDH assay. When measured by LDH assay and expressed as a percentage of cell viability normalized to untreated controls, the reduction in viability of blue-light-induced A2E-loaded ARPE-19 cells was 40.83 ± 1.72% (Figure 4A). To evaluate the effects of *C. tora*, a cell viability assay was performed. A2E was accumulated in RPE cells for 2 weeks and then treated with *C. tora* for 3 days. Lutein, which was used as the positive control, showed a 36.5 ± 3.1% increase in cell viability compared with the A2E blue light group (Figure 4A). We observed the inhibition of cytotoxicity toward blue-light-induced A2E-loaded ARPE-19 cells for *C. tora* fractions. The protective effect of the *n*-hexane fraction was found to be 42.3 ± 1.4%, 44.7 ± 0.8%, 35.3 ± 1.4%, and 33.0 ± 0.9% for the 50, 100, 150, and 200 μg/mL concentrations, respectively (Figure 4B). In the EtOAc, *n*-BuOH, and aqueous fractions, no protection was afforded by all concentrations. Using bioactivity-guided fractionation and isolation techniques, pure chrysophanol and rufbrofusarin were obtained from the *n*-hexane fraction of ECS. The protective effect of chrysophanol was found to be 44.3 ± 2.0%, 47.4 ± 5.7%, 50.9 ± 1.6%, and 52.8 ± 2.1% for the 10, 20, 40, and 80 μM concentrations, respectively (Figure 4C and Figure 3D).

### 2.3. Anti-Apoptotic Effect of C. tora Seed Extract, n-Hexane Fraction, Rufbrofusarin, and Chrysophanol on Expression of Major Apoptosis Factors in Blue-Light-Induced A2E-Loaded ARPE-19 Cells

Here, we determined whether the blue light exposure of A2E-loaded ARPE-19 cells leads to the upregulation of apoptotic genes’ expression. We found that *C. tora* and its components affected the expression of apoptosis factors, such as ABL1, TP53, MAPK8, MAPK9, and MAPK14. To determine whether the mRNA expression of ABL1, TP53, MAPK8, MAPK9, and MAPK14 were activated after the treatment with *C. tora* and its components, the cells were exposed to blue light for 7 min and harvested after 3 h.

ABL1, TP53, MAPK14, and MAPK8 mRNA expressions in the A2E blue light group increased by 89.6%, 92.5%, 71.9%, and 33.5% compared to those in the A2E group, respectively (Figure 5A–D). MAPK9 mRNA expression was increased by 12.3%, but there was no statistical significance (Figure 5E). Treatment of cells with *C. tora* and its *n*-hexane fraction decreased ABL1, TP53, and MAPK14 mRNA levels compared to those in the A2E BL group. Treatment with *C. tora* decreased the mRNA expression by 52.5% in ABL1, 41.1% in TP53, and 17.8% in MAPK14. However, the MAPK8 and MAPK9 mRNA expressions were not decreased by *C. tora*. Chrysophanol possesses superior efficacy, with 82.6%, and 88.4% downregulated ABL1 gene expressions compared to those in the A2E blue light group, respectively. The MAPK14 gene expression in the A2E blue light group increased by 3.3 fold compared to that in the A2E control group, and the treatment of cells with compounds did not reverse the gene expression compared to those in the A2E blue light group. However, chrysophanol inhibited mRNA expression compared to those in the A2E blue light group. MAPK8 mRNA expression was increased by 22.5% in the A2E blue light group; however, MAPK9 mRNA expression was not significantly changed by the treatment of compounds.

## 3. Discussion

In this study, we examined whether *C. tora* extract and its major components (chrysophanol and rubrofusain) have anti-apoptotic effects on blue-light-induced apoptosis. For this experiment, the RPE cell of the retina was used, and after reacting the A2E material with the RPE cell, blue light was formed to induce the photo-oxidation of the A2E material. A2E, a pyridinium bisretinoid, is the main component of drusen in AMD. A2E materials are key photosensitive fluorophores that mediate lipofuscin phototoxicity [22]. With a maximum absorption around 440 nm, A2E and iso-A2E are excited by blue light. The photosensitization of A2E leads to the formation of reactive oxygen species (ROS) and an inhibition of lysozyme’s ability to break down cellular structures for recycling [23,24]. Blue-light-induced A2E oxidation also leads to the formation of A2E oxidative products. Excessive oxidative stress can cause dysfunction in RPE cells and eventually result in cell death by apoptosis. We confirmed that the excessive oxidation of A2E by blue light causes cell death. It could be said that the cell death of RPE by blue light contributes to the onset of AMD. 

First, we treated an extract of *C. tora* with each concentration (50, 100, and 150 ug/mL) to investigate its inhibitory effect on the apoptosis of RPE cells due to A2E photo-oxidation, which we confirmed. Second, each fraction of *C. tora* was evaluated to determine which component of *C. tora* has an anti-apoptotic effect. Among the several fractions of *C. tora*, it was confirmed that the *n*-hexane fraction had the highest inhibitory effect on apoptosis. Chrysophanol and rubrofusain in the *n*-hexane fraction were the main components found using HPLC analysis. Chrysophanol treatments of 10, 20, 40, and 80 uM inhibited apoptosis, while rubrofusarin did not. The anti-apoptotic effect of chrysophanol identified through the results was similar to that of the previously known protective effects of lutein and polyphenol [25,26].

To confirm chrysophanol’s mechanism of apoptosis, the expression of genes involved in the apoptosis pathway were compared and analyzed using RT-PCR. As a result of the analysis, it was confirmed that chrysophanol inhibited the increased expression of ABL1, TP53, MAPK14, and MAPK8 mRNA by photo-oxidation of blue light.

According to previous reports, the programming of cell death involves the activation of caspase-3 and upregulation of bcl-2 [27,28,29]. The ABL1, TP53, and stress kinase pathways have been shown to signal upstream caspase-3 and bcl-2 [11]. ABL1 is a ubiquitously expressed non-receptor tyrosine kinase that is localized at several subcellular sites. ABL1 activity has been associated in the regulation of various responses, such as the cell induction of apoptosis and cycle arrest [30]. 

In human RPE cells, DNA damage induces apoptosis by mechanisms that require a TP53 transcription factor, and ABL1-induced apoptosis is described to be performed by not only TP53-dependent but also TP53-independent mechanisms [31]. The c-Jun *n*-terminal kinases (JNK) and the MAPK14 kinases are the main signaling pathways of the mitogen-activated protein kinases (MAPK) [32]. JNK and MAPK14 are activated by DNA damage, UV irradiation, and oxidative stress.

According to a previous report, the ABL1, TP53, and MAPK9 mRNA expressions of blue-light-induced A2E-loaded ARPE 19 cells were increased compared to those in RPE cells, but MAPK14 mRNA expression did not change [11]. In our blue-light-induced A2E-loaded ARPE-19 cells toxicity model (A2E BL group), the increased expression of ABL1 and TP53 mRNA were significantly inhibited by the treatment with *C. tora* and its active principles, which represented strong evidence that *C. tora* and chrysophanol can be effective natural antioxidants and provide potential chemical interventions for A2E-related diseases, including AMD. From the results on MAPK14 and JNK mRNA expression, it is plausible that *C. tora* and its anthraquinones exerted their anti-apoptotic efficacies mainly through ABL1 and TP53 rather than the MAPK14 and MAPK pathways.

Lai et al. reported that chrysophanol reduced ROS levels in RPE-19 cells exposed to H_2_O_2_ and protected against light-induced retinal damage in animal models [33]. Chrysophanol also protects against cerebral ischemia/reperfusion injury by inhibiting oxidative stress and apoptosis in mice [34]. In addition, chrysophanol can suppress NF-κB/caspase-1 activation during lipopolysaccharide-induced inflammatory responses in mouse peritoneal macrophages [35]. 

In summary, an effective therapeutic agent for AMD has yet to be developed. To provide a practical approach for protecting from AMD caused by excessive exposure to blue light, *C. tora* and chrysophanol were tested for their efficacy in the reduction in RPE-cell-directed cytotoxicity induced by blue light irradiation. *C. tora* and chrysophanol are considered to result in the suppression of apoptosis and oxidative stress reactions. We identified chrysophanol’s potentially valuable anti-apoptotic effects from a medicinal plant, *C. tora*, by protecting A2E-loaded ARPE-19 cells from blue light irradiation. Our results suggested that *C. tora* and chrysophanol may be potentially valuable sources of natural therapeutic agents for the treatment of AMD patients.

## 4. Materials and Methods

### 4.1. Materials

The seeds of *C. tora* were obtained from a domestic Korean market (Kyungdong Crude Drugs Market, Seoul, Republic of Korea) in August 2016. All solvents for extraction, separation and HPLC analysis were purchased from Fisher Scientific Korea (Seoul, Republic of Korea). All-trans-retinal, ethanolamine, and streptomycin/penicillin were purchased from Sigma-Aldrich Co. (St. Louis, MO, USA), and Dulbecco’s Modified Eagle Medium (DMEM) and bovine fatal serum were obtained from Welgene Inc. (Daegu, Republic of Korea). The lactate dehydrogenase (LDH) assay kit was purchased from Daeillab Service Co. (Seoul, Republic of Korea). c-Abl, p53, JNK1, JNK2, and p38 primers were designed by Bionics Inc. (Seoul, Republic of Korea).

### 4.2. Extraction, Fractionation, and Identification

The ground seed of *C. tora* (400 g) was extracted five times with 70% ethanol under reflux (60 °C) for 2 h. Using fractionation guided by bioactivity, the extract was evaporated to dryness under reduced pressure to produce extract. The extract was serially extracted from the filtrate by liquid–liquid extraction (LLE) with *n*-hexane, EtOAc, *n*-BuOH, and H_2_O. *C. tora* seed extracts and compounds were analyzed by high-performance liquid chromatography (HPLC) using an analytical column (4.6 mm × 150 mm, 5 µm, Atlantis^®^ T3, Waters, Ireland). The isolated compound was analyzed by the JMS-T100TD AccuTOF^®^ single-reflection time-of-flight mass spectrometer connected with an electrospray ionization (JEOL Ltd., Tokyo, Japan). All spectra were recorded on a DDS700 MHz liquid and solid NMR spectrometer (Agilent Technologies, Santa Clara, CA, USA) operating at an NMR frequency of 700 MHz. The spectra were attributed to tetramethylsilane (TMS) at 0.00 ppm.

### 4.3. A2E-Loaded ARPE-19 Cell Culture and Irradiation of Blue Light

The synthesis of A2E was performed according to the previous report (Figure 6) [36]. Human adult RPE cells (ARPE-19 cells, American Type Culture Collection, Manassas, VA, USA) were grown in Dulbecco’s Modified Eagle Medium with 10% heat-inactivated fetal bovine serum and streptomycin/penicillin in a humidified atmosphere with a 5% CO_2_ incubator at 37 °C. A2E was added to the culture medium at a concentration of 20 μM, with a total volume of 500 μL per well. A2E was provided for two weeks. To determine the anti-apoptotic effects of *C. tora* extract, its fraction, and compound, the cells that had accumulated A2E were incubated with various concentrations of samples for another 3 days in the absence of A2E. After A2E and sample loading, the cells were exposed to 430 nm light delivered from a light-emitting diode (LED) with a blue light source (420–470 nm, 9.4 mW/cm^2^) for 10 min and incubated for an additional 24 h. Lutein (40 μg/mL) was used as a positive control. A2E, all samples, and lutein were prepared with dimethyl sulfoxide (DMSO). To evaluate the effects of seeds of *C. tora*, A2E, A2E + blue light, and A2E + blue light + samples groups were divided. The design of the experiment in the present study is shown (Figure 7).

### 4.4. Cell Cytotoxicity Assay

Cell cytotoxicity was measured 24 h after 10 min of blue light irradiation via lactate dehydrogenase (LDH) assay. The LDH assays were performed using RPE cell cultures grown to confluence in 24 well plates, and LDH content in cell culture supernatants was measured. The LDH assay kit was purchased from Daeillab Service Co. (Seoul, Republic of Korea). Cell cytotoxicity proceeded according to the protocol. Briefly, 50 μL of cell medium was added to an assay plate; then, 50 μL of assay buffer was added to react for 1 h. After adding 50 μL stop solution, the absorbance value was measured and calculated at 490 nm [37]. A2E, lutein, and samples used in the experiment were dissolved using dimethyl sulfoxide (DMSO), and the final concentration of DMSO was treated at 0.01% or less.

### 4.5. RNA Extraction and Real-Time Polymerase Chain Reaction (RT-PCR)

A2E-loaded ARPE-19 cells grown for 2 weeks and sampled for 3 days were irradiated at 430 nm (7 min) followed by a 2-h incubation. The concentration of the sample used in the experiment was treated at 100 μg/mL for the *C. tora* extracts and *n*-hexane fraction, and 20 μM for rubrofusarin and chrysophanol, respectively. Control groups of A2E-loaded ARPE-19 cells had not been exposed to blue light. Total RNA was extracted using easy-BLUE reagent (iNtRON, Seongnam, Republic of Korea) according to the producer’s protocol. The mRNA was treated for cDNA synthesis using a cDNA synthesis kit (Takara, Tokyo, Japan). After cDNA synthesis, quantitative real-time polymerase chain reaction (RT-PCR) was performed on ABI StepOnePlus Real-Time PCR (Applied Biosystems, Foster City, CA, USA) using the SYBR premix EX Taq one-step kit (Takara). Table 1 shows the primer sequences used. Expression was normalized to GAPDH.

### 4.6. Statistical Analysis

Data were expressed as mean ± SEM. To evaluate the significances of the differences among all groups (A2E, A2E + blue light, and A2E + blue light + samples), one-way analysis of variance (ANOVA) and the Newman–Keuls multiple comparison test were applied using Graph Pad Prism 6statistical software (San Diego, CA, USA); *p* > 0.05 was considered statistically significant.

## Figures and Tables

**Figure 1 ijms-24-06676-f001:**
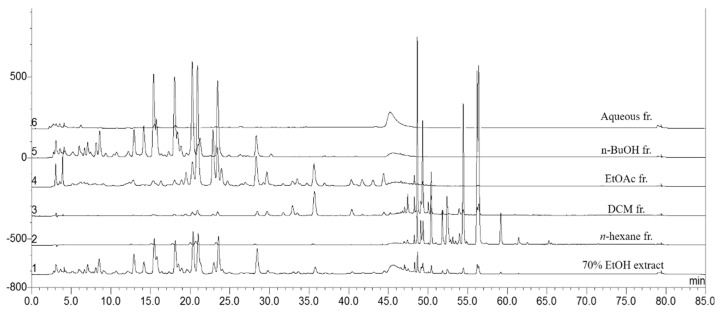
HPLC chromatogram from seed of *Cassia tora* of total extract and its fractionation.

**Figure 2 ijms-24-06676-f002:**
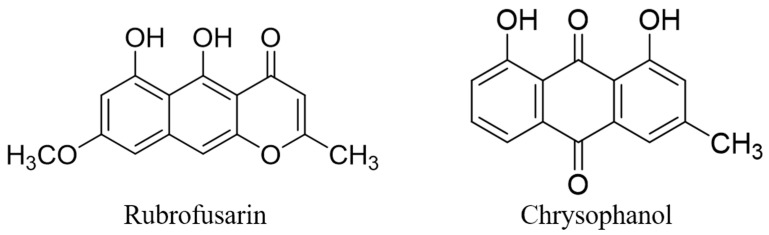
Structures of isolated compounds rubrofusarin and chrysophanol. HPLC, high-performance liquid chromatography.

**Figure 3 ijms-24-06676-f003:**
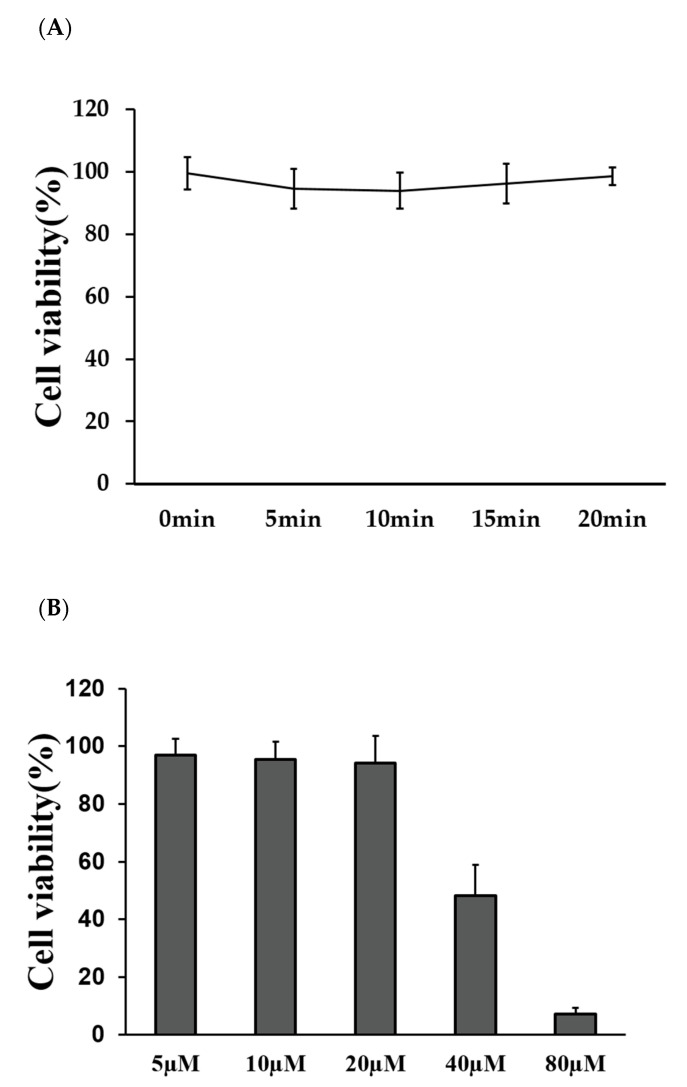
Results of cell cytotoxicity in each group using LDH assay. (**A**) Blue-light-induced cytotoxicity in ARPE-19 cells without A2E, (**B**) the toxicity of A2E in the dark was examined. Cell death was quantified by LDH assay; absorbance value was normalized to untreated control samples (only A2E-loaded ARPE-19 cells). Values are mean ± SEM of three independent experiments performed at different times.

**Figure 4 ijms-24-06676-f004:**
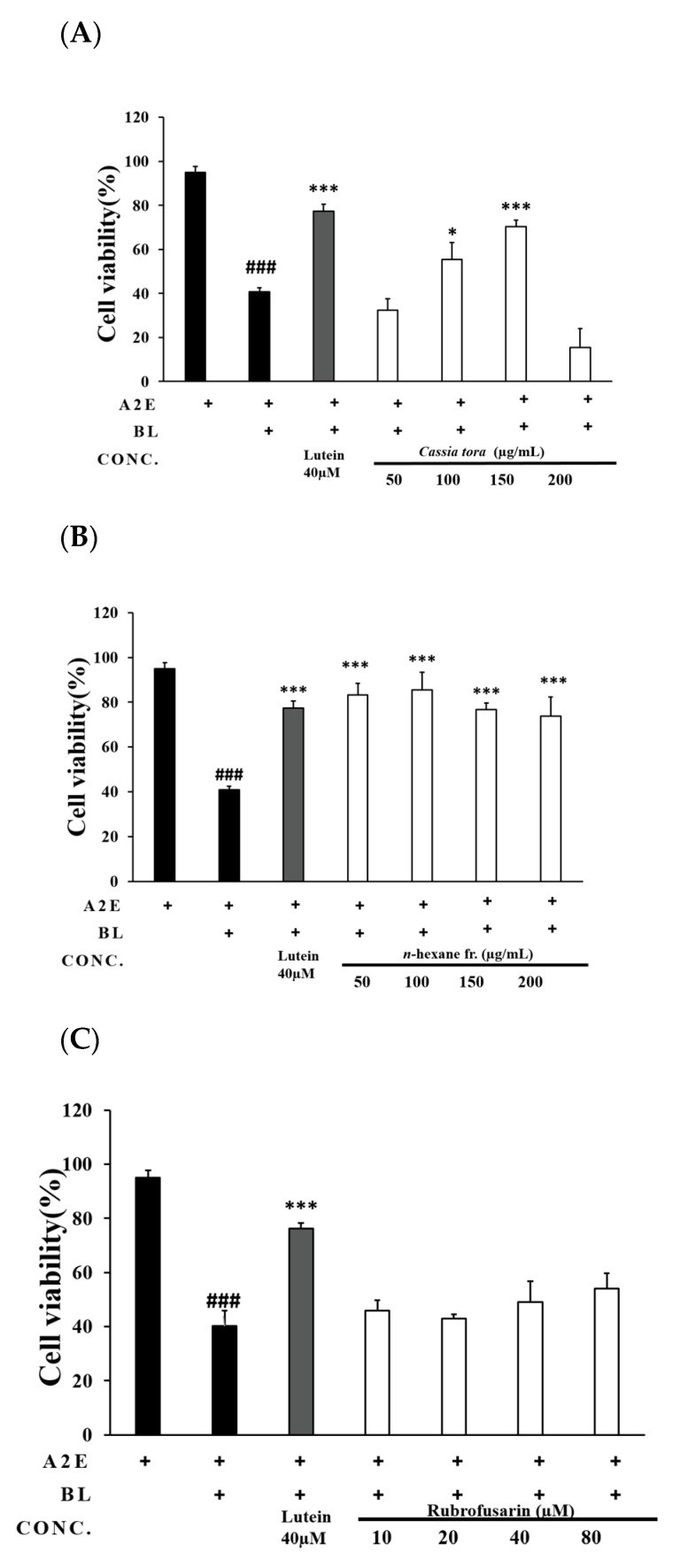
Results of cell cytotoxicity in each group using LDH assay. *Cassia tora* seed extract, fractionation, and compounds inhibit blue-light-induced damage in ARPE-19 cells. (**A**) Cell cytotoxicity of extract, (**B**) *n*-hexane fraction, (**C**) rubrofusarin, and (**D**) chrysophanol. Cell death was quantified by LDH assay; absorbance value was normalized to untreated control samples (only A2E-loaded ARPE-19 cells). Values are mean ± SEM of three independent experiments performed at different times. * *p* < 0.05, and *** *p* < 0.001 vs. A2E blue light group; ### *p* < 0.001, positive control vs. A2E blue light group. LDH, lactate dehydrogenase; SEM, standard error of mean.

**Figure 5 ijms-24-06676-f005:**
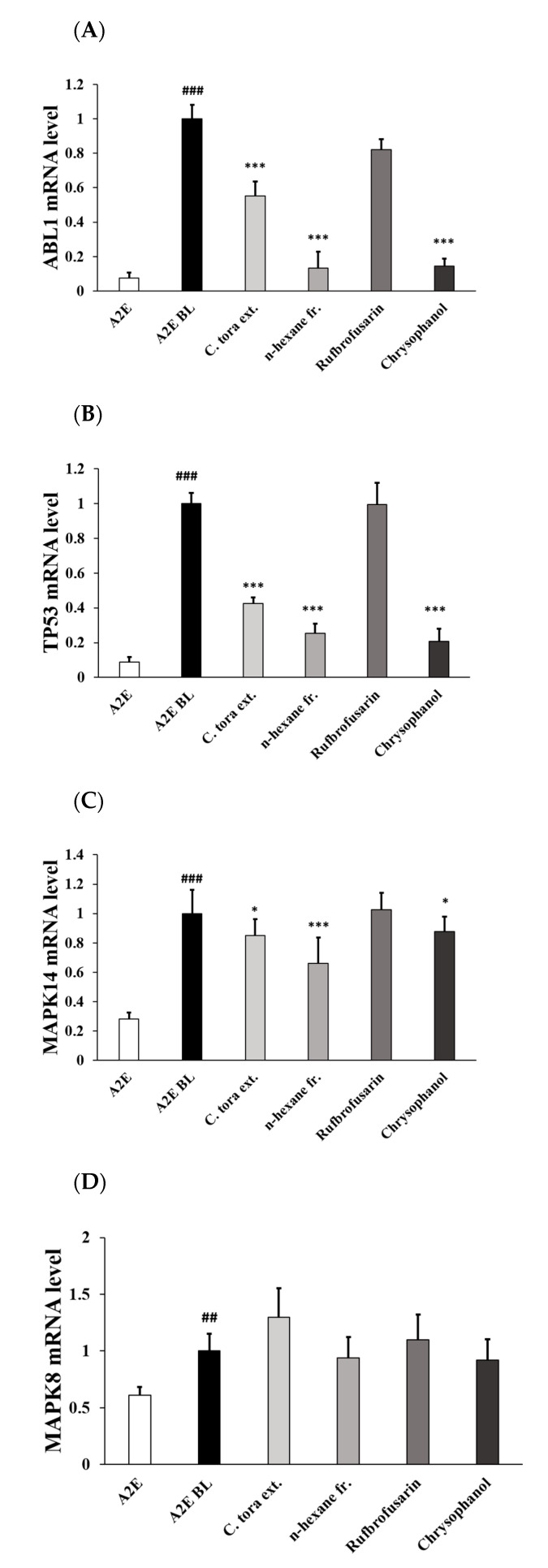
mRNA expressions related to apoptosis pathway in each group using RT-PCR. The effects of *Cassia tora* seed extract, fractionation, and compounds on apoptosis-associated mRNA expression of ABL1 (**A**), TP53 (**B**), MAPK14 (**C**), MAPK8 (**D**), and MAPK9 (**E**) in blue-light-induced A2E-loaded ARPE-19 cells were observed. Values are mean ± SEM of three independent experiments performed at different times. * *p* < 0.05, and *** *p* < 0.001 vs. A2E blue light group; ### *p* < 0.001, and ## *p* < 0.01 positive control vs. A2E blue light group. RT-PCR, real-time polymerase chain reaction; ABL1, ABL proto-oncogene 1, non-receptor tyrosine kinase; TTP53, TTP53 tumor protein TP53; MAPK8, mitogen-activated protein kinase 8; MAPK9, mitogen-activated protein kinase 9; MAPK14, mitogen-activated protein kinase 14; GAPDH, glyceraldehyde-3-phosphate dehydrogenase; SEM, standard error of mean.

**Figure 6 ijms-24-06676-f006:**
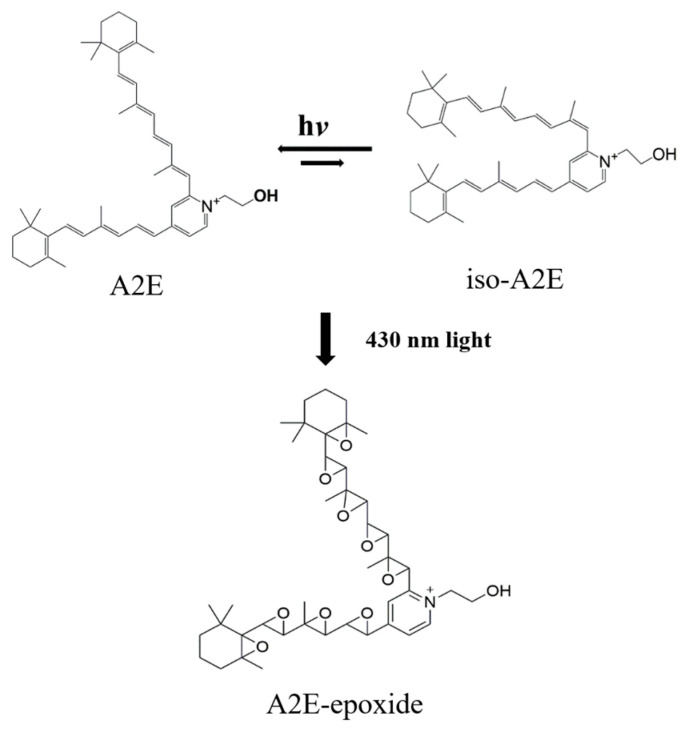
Structure of autofluorescent pigments, A2E, iso-A2E, and A2E-epoxide.

**Figure 7 ijms-24-06676-f007:**
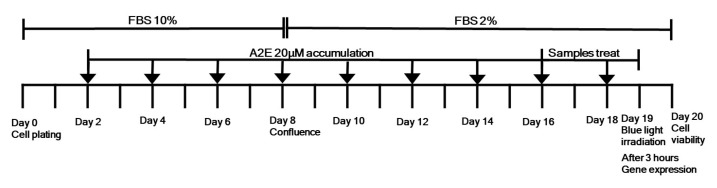
Design of the experiment in the present study.

**Table 1 ijms-24-06676-t001:** Primer sequences used for quantitative RT-PCR analysis.

Genes	Forward (5′–3′)	Reverse (5′–3′)
ABL1	CGGGTCTTAGGCTATAATCAC	CCCTCCCTTCGTATCTCA
TP53	GCAGTCAGAGCCTAGCG	CGCTAGGATCTGACTGC
MAPK8	CTGCCCCCGTATAACTC	CTGCCCCCGTATAACTC
MAPK9	CCTGGGTATGGGCTAC	CGCAGAGCTTCGTCTA
MAPK14	CTCGTTGGAACCCCAG	CATGTGCAAGGGCTTG
GAPDH	CTGAGCTGAACGGGAAG	GGGTGTCGCTGTTGAA

RT-PCR, real-time polymerase chain reaction; ABL1, ABL proto-oncogene 1, non-receptor tyrosine kinase; TTP53, TTP53 tumor protein TP53; MAPK8, mitogen-activated protein kinase 8; MAPK9, mitogen-activated protein kinase 9; MAPK14, mitogen-activated protein kinase 14; GAPDH, glyceraldehyde-3-phosphate dehydrogenase.

## Data Availability

The data will be available by corresponding authors upon genuine request.

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
