# Peer review of "Anti-Apoptotic Effect of Chrysophanol Isolated from Cassia tora Seed Extract on Blue-Light-Induced A2E-Loaded Human Retinal Pigment Epithelial Cells"

_ijms, 2023, doi:10.3390/ijms24076676_

Round 1

Reviewer 1 Report (Previous Reviewer 2)

Dear authors, 

You have made some improvement to the paper but still I have some comment collected in a file attached. 

First, you have to update the introduction with the latest work on C. Tora and chrysophanol. Then you have to point out what's your objective compared to what has already been done.

Mat and Meth : why didn't you use DMSO treated cells as control. 

Third in the discussion you have to comment more your work compared to what was already known. 

Author Response

Manuscript No.: Ijms-2300166

â–  Authors: Su Kang Kim et al.

â–  Title: " Anti-apoptotic Effect of Chrysophanol Isolated from Cassia Tora Seeds Extract on Blue Light-Induced A2E-Loaded Human Retinal Pigment Epithelial Cells"

â–  Type of manuscript: Article

â–  Submitted to: International Journal of Molecular Sciences

  Thank you very much for considering our manuscript for publication. Your suggestions were very helpful to us, and we have incorporated those points into our revised manuscript.

Reviewer 2 Report (Previous Reviewer 1)

Authors addressed my previous comments. Article can be accepted for the publication.

Author Response

Manuscript No.: Ijms-2300166

â–  Authors: Su Kang Kim et al.

â–  Title: " Anti-apoptotic Effect of Chrysophanol Isolated from Cassia Tora Seeds Extract on Blue Light-Induced A2E-Loaded Human Retinal Pigment Epithelial Cells"

â–  Type of manuscript: Article

â–  Submitted to: International Journal of Molecular Sciences

  Thank you very much for considering our manuscript for publication. Your suggestions were very helpful to us, and we have incorporated those points into our revised manuscript.

Round 2

Reviewer 1 Report (Previous Reviewer 2)

Dear Author, 

thank you for answering all the request and making the changes that have been suggested.

This manuscript is a resubmission of an earlier submission. The following is a list of the peer review reports and author responses from that submission.

Round 1

Reviewer 1 Report

Major comments:

1.     Enough evidence should be presented that apoptosis inhibition is beneficial long-term. Are damaged but alive cells better than dead cells? If we inhibit apoptosis more DNA mutation load will accumulate in RPE cells and we don’t know the consequences of this.

2.     Statistics. It is not obvious why SEM was used instead of SD. Usually that makes graphics more beautiful. I would recommend to thorough double-check the statistics and raw data for the level of statistical significance, especially in the light that the substance might eventually go to the market.

3. Design of experiment would benefit if some schematic visualization added

Minor comments:

1.     Line 265: typo: RPE should be instead of REP.

2.     Line 269: typo: treatment of C. tora – should be treatment with C. tora instead

3.     Line 269: typo: ) should be instead of ))

4.     Line 274-275 typo: treatments of chrysophanol – incorrect. treatments with chrysophanol - correct. Please check with the native speaker and grammar check.

5.     Line 294: typo: m RNA – should be written with no space: mRNA.

6.     Line 309: “practical approach for protecting AMD caused by excessive exposure to blue light…” It is advised to rephrase the sentence. We should not protect AMD. We should protect patient’s retina instead. Or need to insert “practical approach for protecting FROM AMD caused by excessive exposure to blue light…”

Author Response

Date: Feb. 16, 2023

Manuscript No.: Ijms-2162388

â–  Authors: Su Kang Kim et al.

â–  Title: " Anti-apoptotic Effect of Chrysophanol Isolated from Cassia Tora Seeds Extract on Blue Light-Induced A2E-Loaded Human Retinal Pigment Epithelial Cells"

â–  Type of manuscript: Article

â–  Submitted to: International Journal of Molecular Sciences

  Thank you very much for considering our manuscript for publication. Your suggestions were very helpful to us, and we have incorporated those points into our revised manuscript.

Reviewer 2 Report

The author have studied the anti-apoptotic effect of Cassia Tora extracts on Blue light-induced damage in RPE cells loaded with A2E. They quantified damage with LDH assay and RNA expression. Although the results are interresting, the paper has to be improved.

Mat and Meth : You have to be more precise in the material and methods used

             What kind of RPE cells has been used ? Is it ARPE19 ?

How were the cells seeded ? what kind of well -plates (96, 24 ?) at which concentration ?

dont' you let the cells grow before use (before providing A2E) ? If so for how long ?

It is not clear what are the control and how the treatment are prepared (soluble in water ? or do you use another vehicule ?)

It is not clear what are the different groups : line 131 (Statistical analysis) you talk about A2E + Blue light , A2E + Blue Light + Samples, A2E and A2E-free which are not explained in Mat and Meth,

then line 176à untreated control

Line 179 : Lutein is not mentionned in Mat and Meth, where does it come from and how do you prepare

Line 185 : A2E BL is not mentionned in Mat and Meth.

Line 141 : 2 coumpounds isolated from n-BuOH while line 188 it is said « n-hexane fraction ».

Why is the timing for RNA different from the LDH assay : 7 min of light followed by 3h for RNA and 10 min of light followed by 24h for LDH

Results :

Line 137 « The extracted seed of C. tora was evaporated to dryness under reduced pressure to give 70% ethanol extract : if the extract are evaporated to dryness they is no ethanol ?? I supposed then that you solubilize in 70% ethanol ? if so then a control should be 70% ethanol ?

Line 153 and 164 « that was confirmed by rubrofusarin » and « that was confirmed by chrysophanol 26255810 » :  do you mean « that was confirmed as being rubrofusarin » that was confirmed to be chrysophanol?

Line 177-178 : not clear

Line 179 Lutein at which concentation and how did you solubilized ?

Line 181 : C. tora were found to be a 14.5% and 29.5% à this sentence does not mean anything

Line 187 : Using bioactivity-guided fractionationà this is not mentioned as it in Mat and Meth (you have to homogenise)

You have to give the variation of the values as X ± Y %

 Figure 4A : Are the treated group significantly different between each other?

What is the equivalence between the concentration of the isolated coumpound compared to the amount in  the extract tested?

 Fig 5D : Is  n-hexane fraction really significantly different from A2E BL?

Discussion : The discussion need to be improved

Line 274-275 : contradictory points in the same sentence.

Line 281 : « The ABL1, TP53 and stress kinase pathway had been 281 shown to upstream signaling caspase-3 and bcl-2 »  the paper cited (28) from Sparrow and Cai do not study ABL1, TP53 or stress kinase pathway.

Line 284 : «… such as cell induction of apoptosis and cycle arrest (29) « : however, the paper from Sawyers et al., 1994 shows that  c-abl acts as a negative regulator of cell growth but they do not show that it induces apoptosis.

Line 288 : the paper cited (30) from Yang et al, 2014 does not show that « In human RPE cell, DNA damage induces apoptosis by mechanisms that require TP53 transcription factor, and ABL1 induced apoptosis is described to be performed by not only TP53-dependent but also TP53-independent mechanisms »

Author Response

(The authors gave the same response as above.)

Reviewer 3 Report

This manuscript describes multiple experiments evaluating C. tora seed extract and related compounds on preventing apoptosis in RPE cells that have accumulated A2E and been exposed to blue light treatment. The authors show the seed extract and some of its related compounds can prevent RPE cell death through cell viability and gene expression assays. However, a major flaw in this work is the lack of proper controls. There is no data on RPE cell cultures with no exposure to A2E. It has been shown by previous studies that A2E can induce cell death in RPE cells within 3 to 6 hours after exposure (PMID: 32290199). Therefore, this is a critical control for these experiments. Also, the solvents used to dissolve the C.tora seed extract and its related compounds were not incubated with controls in these experiments, thus there is the possibility that the solvents used in these experiments could be causing these results. Other minor issues with the manuscript include a lack of experimental details (i.e. what are the concentrations of the seed extract/related compounds in Figure 5 and why were these genes chosen), no histology of the ARPE-19 cells was performed, no validation of the amount of A2E in the RPE cells, and no protein validation of the key findings in this study. 

Author Response

(The authors gave the same response as above.)

Round 2

Reviewer 1 Report

Authors addressed y previous comments. The article can be accepted after minor revision. 

Minor comment: line 333 FROM should not be in capital letters. from - is correct.

Reviewer 3 Report

This reviewer appreciates the authors’ revisions to their manuscript. The addition of more experimental details has helped this reviewer to understand their study and results more than the previous version of their draft. 

Major concerns:

Although the authors claim to not see any effect of A2E or blue light on RPE cell viability, this data should be included and referenced in the results section. 

Were all treatments for their cell culture experiments performed at the same time? It is unclear from the manuscript and their rebuttal. This reviewer is assuming that these experiments were performed at the same time since the control groups are the same throughout Figure 5.

Can the authors clarify what they meant by three experiments? Is this meaning a triplicate of cell culture wells with the same treatment? Or three independent experiments performed at different times?

It is still unclear the amount of the compound used for the treatments for Figure 6. 

This reviewer recognizes that the extraction of the compound is an important process and that the authors devoted their time and efforts in completing this work. However, this paper is not focusing on the extraction of the C. toro seed extract. This paper is on the effect of the C. toro seed extract on apoptosis of ARPE-19 cells (as described by the title of their paper). There are only two figures presented in this manuscript that evaluate apoptosis in ARPE-19 cells. There needed to be more work that established C. toro seed extract prevents apoptosis in ARPE-19 cells (i.e. protein work, staining of cells, etc). The authors describe various pathways involved with apoptosis in RPE cells and these pathways are a good place to start. 
